# A New Possible Cut-Off of Cytokeratin 19 mRNA Copy Number by OSNA in the Sentinel Node of Breast Cancer Patients to Avoid Unnecessary Axillary Dissection: A 10-Year Experience in a Tertiary Breast Unit

**DOI:** 10.3390/cancers14143384

**Published:** 2022-07-12

**Authors:** Giovanni Tomasicchio, Mauro Giuseppe Mastropasqua, Arcangelo Picciariello, Alda Elena Montanaro, Daniela Signorile, Alfredo Cirilli, Clelia Punzo

**Affiliations:** 1Department of Emergency and Organ Transplantation, School of Medicine, University of Bari “Aldo Moro”, Piazza G Cesare, 11, 70124 Bari, Italy; arcangelo.picciariello@uniba.it (A.P.); clelia.punzo@uniba.it (C.P.); 2Breast Unit Surgery, Azienda Ospedaliera Universitaria Policlinico Bari, Piazza G Cesare, 11, 70124 Bari, Italy; a.montanaro13@libero.it (A.E.M.); daniela.signorile@policlinico.ba.it (D.S.); cirilli.alfredo@libero.it (A.C.)

**Keywords:** breast cancer, breast pathology, breast surgery, axillary lymph nodes, sentinel lymph node, one-step nucleic acid amplification, cytokeratin 19, RNA, messenger, retrospective studies, sensitivity and specificity

## Abstract

**Simple Summary:**

This manuscript aims to investigate the features of patients with metastatic sentinel lymph node (SLN), evaluated by OSNA, and to predict which patients have a high risk of positive ALND. The finding of the present study suggests a new cut-off of CK19 mRNA copy number in the sentinel lymph node useful to personalize surgical treatments and avoid unnecessary axillary surgical treatments.

**Abstract:**

(1) Background: The main discriminant in breast cancer prognosis is axillary lymph node status. In a select cohort of patients, axillary lymph node dissection (ALND) may be safely spared. This study aimed to determine a new possible cut-off of cytokeratin (CK) 19 mRNA copy number in the SLN to predict cases at high risk of positive ALND. (2) Methods: Clinical records of 1339 patients were retrospectively reviewed and were separated into two groups according to the axillary status (negative: ALNs− and positive ALNs+). Receiver operative characteristic (ROC) curves were used to identify a new optimal cut-off of CK19 mRNA copy number in SLN; (3) Results: Large tumor size and high grade were found mostly in ALNs+. Results from the ROC analyses, with an AUC of 82.1%, identified a new cut-off (9150 CK19 mRNA copies) showing 94% sensitivity, 67.3% specificity, 61.2% positive, and 95.3% negative predictive values; (4) OSNA remains the most-important intra-operative tool to identify patients who can benefit from ALND but with the traditional cut-off, many patients undergo needless ALND. The results of the present study suggest a new cut-off helpful to personalize surgical treatment and avoid unnecessary invasive procedures.

## 1. Introduction

The main discriminant in breast cancer prognosis is axillary lymph node status [1]. In fact, the recommended procedure for the axillary staging of early breast cancer with clinically negative axilla is sentinel lymph node (SLN) biopsy that has replaced axillary lymph node dissection (ALND) [2,3]. Currently, the intra-operative histological examination of frozen section or serial section slides allows one to detect SLN metastases. Nevertheless, the histopathological examination may lead to a high rate of false-negative results since it cannot measure total metastatic volume because only a small portion of the sentinel lymph node can be analyzed. Furthermore, the wide assortment of techniques currently adopted impedes the standardization of this procedure [4,5]. Since 2007, the One-Step Nucleic acid Amplification (OSNA) assay (Sysmex, Kobe, Japan) has been validated as a semi-automatic, standardized, reproducible, and intra-operative procedure for SLN examination [6]. In fact, the OSNA assay has been shown to be more sensitive and to detect more sentinel lymph node metastases with an increased lymph node-positive rate of 30% compared to frozen histological examination [4]. This rapid molecular detection procedure analyses homogenized SLN by rapid real-time amplification and quantifies Cytokeratin 19 (CK19) mRNA copy numbers, which are expressed in most breast cancer cells [7]. Tsujimoto et al. defined cut-off values to discriminate negative node (less than 250 CK19 mRNA copies/µL) from micro-metastases (250–5000 CK19 mRNA copies/µL) or macro-metastases (more than 5000 CK19 mRNA copies/µL) [6].

The American College of Surgeon Oncology Group (ACOSOG) Z0011 trial has defined a selected cohort of patients with one or two positive SLNs in which ALND may be safely avoided [8]. In fact, several studies have recently proposed different nomograms to spare a complete ALND in patients with a low risk of metastasis in non-sentinel axillary nodes [9,10]. Nevertheless, many of them are built on histological information that cannot be used intraoperatively and most variables considered are subjective, difficult to reproduce, and available only after surgery [11].

This retrospective observational single-center study aims to explore the features of patients with macro-metastases at SLN evaluated by OSNA, with or without positive non-sentinel axillary nodes at ALND, and to determine a new possible cut-off of CK19 mRNA copy number in the SLN to predict cases at high risk of having a positive ALND.

## 2. Materials and Methods

### 2.1. Patients

A retrospective observational study was carried out using a prospectively maintained database of patients affected by breast cancer who underwent surgery in a tertiary Breast Unit between January 2008 and December 2021.

All patients underwent physical examination, ultrasound, and mammography as part of their diagnostic work-up. We included in this study patients with breast cancer cT1-3 and with negative (cN0) clinical and ultrasound axillary examination who underwent intraoperative SLN assessment by OSNA. We excluded patients who had received neoadjuvant therapy or previous ipsilateral breast or axillary surgery.

Patients with macro-metastasis at SLN were divided into two groups: the group of patients with negative axillary lymph nodes (ALNs−) at ALND and the other group with positive ALNs+ at ALND.

In all patients, morphological characteristics such as histologic types and grades were evaluated. Estrogen receptor (ER) status, Progesterone receptor (PgR) status, HER2 status, and Ki-67 were evaluated on formalin-fixed and paraffin-embedded tissue from core needle biopsy of the primary tumor before breast surgery. We defined cases as ER and PgR positive when at least 1% of tumor cells were stained by immunohistochemistry (IHC) according to the College of American Pathologists (CAP) criteria [12], whereas HER2 IHC positivity was defined according to ASCO/CAP 2018 guidelines [13]. Ki-67 was recorded as a percentage of immunoreactive nuclei in at least 2000 neoplastic cells. A cut-off of 20% was considered to classify cases as high Ki-67 expressing and low Ki-67 expressing tumors, associated with a more aggressive behavior [14].

Using immunohistochemistry for ER, PgR, HER2, and the two-tiered Ki-67 scale, as a surrogate for molecular classification of breast cancer, four intrinsic molecular breast cancer surrogate subtypes were considered: Luminal A, Luminal B, HER 2-enriched, and Basal-like (Triple Negative) [15].

Sentinel lymph node biopsy was performed depending on the pre-operative axillary lymph node ultrasound assessment, along with the imaging tumor size and negative cytology for palpable nodes. In cases of positive OSNA assay (more than 5000 CK19 mRNA copies/µL), ALND was performed in the same surgical session. Tumors were graded according to the Nottingham system and staged according to Union Internationale Contre le Cancer (UICC), tumor-node-metastasis (TNM) system criteria [16].

Written informed consent was obtained from all patients before surgical procedures.

### 2.2. Intraoperative SLN Analysis with OSNA Assay

SLN was localized with a peri-areolar injection using technetium 99 m-labeled, nano-sized, human serum albumin colloids the day before surgery, followed by lymphoscintigraphy 1–3 h later. SLNs were identified, before the primary tumor surgery, using a hand-held gamma-probe, then isolated, and finally, the perinodal fat was removed.

SLNs were sent on ice to Pathology Section and subsequently weighed, measured, and analyzed. Whole SLNs were homogenized with 4 mL of a lysis buffer solution (Lynorhag, Sysmex, Kobe, Japan) and centrifuged at 10,000× *g* for 1 min at room temperature. CK 19 mRNA detection was assessed using reverse transcription-LAMP (Loop-mediated isothermal Amplification) with the RD100i (Sysmex) gene amplification detector.

According to Tsujimoto et al., a copy number > 5000/µL was considered the cutoff predictive of SLN macro-metastases, while between value 250–5000/µL was predictive of SLN micro-metastases and less than 250/µL as the absence of tumor cells [6].

In patients with more than one SLN positive, only the SLN with the highest number of copies was considered for each patient.

The surgeons instantly received OSNA results by phone, usually within 20–30 min. ALND was performed only on patients with macro-metastases, whereas in patients with micro-metastases or with negative results, no further axillary surgery was done [17,18].

### 2.3. Statistical Analysis

All data were reported as median and interquartile ranges. Descriptive data were recorded as numbers and percentages. Comparisons of categorical variables were performed by the Chi-square and Fisher’s Exact test, where appropriate. Comparisons between groups were made by the Mann–Whitney U test. A *p*-value lower than 0.05 was considered statistically significant. Receiver operative characteristics (ROC) curves and Youden’s index were used to identify an optimal cut-off value with the highest sensitivity and specificity. Across various cut-off points, Youden’s index maximized the difference between sensitivity and specificity and between real-positive and false-positive subjects. The optimal cut-off value was calculated. The AUC (area under the curve) was used to summarize the overall diagnostic accuracy of the test. A value of 50% was considered “not discriminating,” a value of 70–80% was considered “acceptable,” a value of 80–90% “excellent, and more than 90% outstanding.

Statistical analysis was carried out using RStudio (R version 4.0.3 (10 October 2020) Copyright © 2020 The R Foundation for Statistical Computing).

## 3. Results

Out of 3590 European patients who underwent surgery between January 2008 and December 2021, 1339 were evaluated intraoperatively for the axillary status by SLN valuation by OSNA and were included in our study according to the study selection criteria. Among the 1339 patients, 1011 (75.5%) patients had a negative SLN and 328 had positive SLN: 126 (9.5%) had micro-metastases and 202 (15%) had macro-metastases. Of the 202 patients with macro-metastases in the SLN, 86 (42.5%) patients had no other positive ALNs at ALND, and the remaining 116 (57.5%) had one or more positive ALNs at ALND.

There were no significant differences between the two groups with regard to the histotype of the primary tumor. The group without positive ALNs had a significantly higher median age compared to the other group (56.5 (IQR, 48.7–66) vs. 52 (IQR, 48–61) respectively, *p* < 0.05), and significant differences regarding tumor size and grading of the primary tumor (*p* < 0.05). A tumor size ≥ 2 cm (T2-3) and high-grade tumors (G3) were found mostly in the group with one or more positive ALNs.

Table 1 summarizes the histopathological characteristics of the primary tumors of the two groups.

The mean age, the percentage of histotypes, hormonal status, and HER2 expression were in line with the literature data. There were statistically significant differences between the two groups regarding the pTNM and the grade, in accordance with the literature data that show a higher likelihood of metastasizing for larger and high-grade tumors.

We recorded no differences between the two groups concerning dimension and SLNs weight, but there was a significant difference in the number of CK19 mRNA copies. It was higher in group with one or more positive ALNs (49,500 (IQR, 11,625–179,700) vs. 89,000 (IQR, 20,000–443,350), *p*-value < 0.05). The median number of ALNs removed at ALND was 19 (IQR, 17–24) in the group with a single positive SLN vs. 21 (IQR, 17.75–24.25) in the group with a median of 3 (IQR, 2–5) positive ALNs (Table 2).

Results from the ROC analyses, with an AUC of 82.1%, identified a cut-off equal to 9150 CK19 mRNA copies. For this analysis, we considered the maximal CK19 mRNA copy number of all the 328 SLN with macro- and micro-metastases. Therefore, two groups were identified. In the group of patients with <9150 CK19 mRNA copies, only 5 patients had positive ALNs at ALND. All of them had only 2 positive ALNs at ALND. Furthermore, these five patients were characterized by a tumor size > 2 cm with a tumor grade ≥ 2. Three patients had invasive ductal carcinoma, the remaining had invasive lobular carcinoma and mucinous carcinoma. The new cut-off showed 94% sensitivity and 67.3% specificity when differentiating patients without positive ALNs vs. those with positive ALNs at ALND (Figure 1).

Positive and negative predictive values of this new cut-off were 61.2% and 95.3%, respectively.

## 4. Discussion

The One-Step-Nuclear-Acid amplification has been reported, over the years, as a standardized intraoperative technique, avoiding sampling errors and second-time surgeries [19]. However, the clinical implication of a positive SLN has been modified in recent years. The ACOSOG Z0011 trial revealed that ALND could be avoided in patients with T1-2 breast cancer and one or two positive SLNs, who were treated with breast-conserving surgery, adjuvant systemic therapy, and whole-breast irradiation [8]. The European Organization for Research and Treatment of Cancer (EORTC) in the AMAROS trial demonstrated the non-inferiority of axillary radiotherapy compared to ALND in early breast cancer, with reduced morbidity [20]. Nevertheless, the applicability of these trials is limited to a selected cohort of patients and to date, ALND remains a common practice in patients with positive SLN. However, ALND is associated with lower quality of life, elevated incidence of arm swelling, lymphedema, sensory morbidity, and an increase in health care costs compared to sentinel lymph node biopsy [21]. For these reasons, it is mandatory to identify preoperative and clinically relevant risk factors to avoid ALND in breast cancer patients.

Recent studies indicate that breast cancer molecular subtypes are closely related to axillary nodal status [22,23]. Luminal B and HER2-enriched cancers are more commonly related to no-sentinel lymph node (NSLN) metastasis as compared to Luminal A and Triple Negative breast cancer [23]. These studies included patients with and without ALN involvement, without delineating a relationship between molecular subtype and NSLN status in positive SLN patients. Interestingly, in our analyses, we did not find any statistical relationship between molecular subtypes, receptor status, HER2 status, Ki-67% and the risk of NSLN metastasis in patients with positive SLN. A recent study [24] demonstrated that Luminal B and HER2 overexpression were independent and significant predictors of NSLN metastasis versus Luminal A in patients with positive SLN. However, in their study, the SLN was analyzed with a frozen technique, with a possible false-negative rate from 26 to 46% when compared to the OSNA assay [4].

Degnim et al. identified, in their meta-analysis, a >2-fold increase in the risk of NSLN metastasis in the presence of one of the following five characteristics: size of primary tumor > 2 cm; size of metastatic focus in SLN > 2 mm; extra-nodal extension in SLN; more than one positive SLN; and peritumoral lympho-vascular invasion in the primary tumor [25]. Tumor size and grading have largely been described as important predictors of NSLN metastasis [24,26,27]; in line with our results, tumor size > 2 cm and high tumor grade were major risk aspects of NSLN metastasis (*p* < 0.05). The role of the size of the metastatic focus and extra-nodal invasion in the SLN was changed with the introduction of the OSNA assay.

Different studies demonstrated a correlation of extracapsular extension with the size of metastatic focus in SLN and a strong association with the likelihood of positive NSLN [25,28]. With OSNA assay, these two SLN characteristics can no longer be investigated because the rapid molecular detection procedure analyzed a homogenized SLN for the quantification of Cytokeratin 19 mRNA copy numbers.

Our study evaluated the characteristics of SLN without finding any statistical correlation between dimension and weight of positive SLN with the risk of NSLN metastasis. Meanwhile, there was a statistical difference between the median number of SLN CK19 mRNA copies. Patients with NSLN metastasis had a higher copy number compared to those without positive ALNs (49,500/µL vs. 89,000/µL, *p* < 0.05).

In this study, a risk analysis of positive ALN in correlation with CK19 mRNA copies in SLN detected by OSNA was carried out to overcome the histological distinction between micro- ad macro-metastases in SLN. A ROC curve was constructed to establish the best cut-off value to provide high specificity and sensitivity, identifying a cut-off equal to 9150 CK19 mRNA copies. Predictive cut-offs of CK19 mRNA copies have already been investigated in literature [29,30,31] and some authors considered the total tumor load (TTL), defined as the amount of CK19 mRNA copies in all positive SLNs [11,32,33,34]. The difference between cut-offs across studies that considered TTL could be explained by the clinical variability of each center. Moreover, TTL is strictly linked to the number of SLNs removed and analyzed. By contrast, in our study, we considered the positive SLN with the maximal copy number to overcome this limitation. Only Heilmann et al. [31] obtained their cut-off by separate 1 mm central slides for histology and the rest for OSNA, but it does not represent the actual copy number of CK19 mRNA of the whole lymph node. The threshold value of 9150 CK19 mRNA copies found in our analysis could be considered too high to allow the identification of patients with positive ALND. However, when compared with the threshold found in the literature, our cut-off with AUC of 82.1%, 94% sensitivity, 67.3% specificity, PPV of 61.2%, and NPV of 95.3% had the highest specificity in identifying patients who do not need ALND.

Furthermore, it is essential to emphasize that our cut-off was identified in a large series of breast cancer patients recruited in the last 10 years. All the studies found in the literature base their cut-off on a smaller cohort of patients during a limited period of time. Only the cut-off of Peg et al. [32] and Espinosa-Bravo et al. [34] was found with a multicentric analysis; all the remaining were single-center studies (Table 3).

The limitation of the study is represented by its retrospective type, which opens it to possible selection bias. A future prospective, multicenter randomized controlled study is necessary to determine whether this cut-off may have a clinical impact on locoregional recurrence and overall survival.

## 5. Conclusions

Preoperative tumor grade and size are traditionally considered relevant prognostic factors of axillary lymph node involvement. Moreover, OSNA remains the most-important intra-operative prognostic tool for the decision-making process for axilla surgery. Nevertheless, using the traditional cut-off, many patients undergo ALND without positive NSLD, with all the possible complications.

The finding of the present study suggests a new cut-off to personalize surgical treatments and avoid unnecessary invasive procedures. The proposed cut-off value of 9150 CK19 mRNA copies could be used in patients with a tumor size < 2 cm (T2) and with a lower grading (G1) to avoid unnecessary axillary surgical treatment, ensure a safe patient outcome, and minimize morbidity.

## Figures and Tables

**Figure 1 cancers-14-03384-f001:**
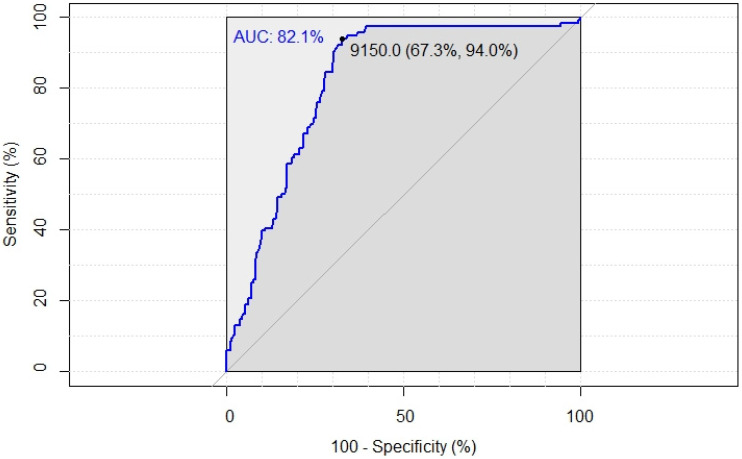
ROC curve. The new cut-off of cytokeratin 19 mRNA copy number was obtained by ROC curve.

**Table 1 cancers-14-03384-t001:** Pathological characteristics of primary tumors of the 2 groups.

Characteristics	No ALN+ (*n* = 86)	≥1 ALN+ (*n* = 116)	*p*-Value
**Age**	51 (49–55)	48 (47–53)	0.41
**BMI (Kg/m^2^)**	25 (22–27)	26 (24–28)	0.33
**Surgical treatment**			
Conservative	72 (84%)	95 (82%)	0.8
Mastectomy	14 (16%)	21 (18%)	0.8
**Histological type**			
Invasive Ductal carcinoma	71 (82.5%)	88 (75.8%)	0.32
Invasive Lobular carcinoma	9 (10.5%)	19 (16.4%)	0.31
Invasive Ductal and Lobular carcinoma	3 (3.5%)	6 (5.2%)	0.7
Ductal Carcinoma In Situ	0	2 (1.7%)	0.5
Other	3 (3.5%)	1 (0.9%)	0.3
**Tumor size**			
pT1a	2 (2.3%)	1 (0.9%)	0.5
pT1b	20 (23.3%)	12 (10.2%)	* <0.05
pT1c	40 (46.5%)	37 (32%)	* <0.05
pT2	23 (26.7%)	52 (44.8%)	* <0.05
pT3	1 (1.2%)	14 (12.1%)	* <0.005
**Grading**			
1	12 (14%)	5 (4%)	* <0.05
2	45 (52%)	46 (40%)	0.09
3	29 (34%)	65 (56%)	* <0.005
**Estrogen Receptor status**			
Positive	74 (86%)	95 (82%)	0.5
Negative	12 (14%)	21 (18%)	0.5
**Progesterone Receptor status**			
Positive	47 (54.6%)	61 (52.5%)	0.8
Negative	39 (45.4%)	55 (47.5%)	0.8
**HER2 expression by IHC**			
Positive	28 (32.5%)	37 (32%)	1
Negative	58 (67.5%)	79 (68%)	1
**Ki67%**			
<20%	47 (54.6%)	68 (58.6%)	0.6
≥20%	39 (45.4%)	48 (41.4%)	0.6
**Surrogate molecular subtypes**			
Luminal A	30 (35%)	46 (40%)	0.5
Luminal B	47(54.6%)	56 (48%)	0.4
HER2 enriched	5 (5.8%)	7 (6%)	1
Triple-negative	4 (4.6%)	7 (6%)	0.7

Histopathological characteristics of the primary tumor of patients with macro-metastasis at SLN with or without positive axillary lymph nodes (ALNs) at ALND. Descriptive data are expressed as percentages (*: statistically significant).

**Table 2 cancers-14-03384-t002:** Characteristics of SNLs of the 2 groups.

Characteristics	No ALN+ (*n* = 86)	≥1 ALN+ (*n* = 116)	*p*-Value
SLN size (main)	1.5 cm (1.2–2)	1.5 cm (1.2–1.9)	0.9
SLN weight	0.58 gm (0.4–0.9)	0.6 gm (0.4–0.9)	0.2
SLN CK19 mRNA copies	49,500 (range 11,625–179,700)	89,000 (range 20,000–443,350)	* <0.05
LNs removed at ALND	19 (17–24)	21 (17.75–24.25)	0.3
LNs positive at ALND	0	3 (2–5)	* <0.005

Characteristics of positive SLN in patients with or without positive axillary lymph nodes (ALNs) at ALND. Data were expressed as median and interquartile ranges (*: statistically significant).

**Table 3 cancers-14-03384-t003:** Comparison of our OSNA cut-off with other alternatives from literature.

Study	N. of Pts	Cut-Off (Copies/µL)	Years	AUC (%)	Se (%)	Sp (%)	PPV (%)	NPV (%)	Method
Present study	1339 202 ++ 126 +	9150	13	82.1	94	67.3	61.2	95.3	Maximalcopy
Tsujimoto et al. [6]		5000			78.6	56.8	44.8	85.6	Maximal copy
Deambrogio et al. [29]	1296 117 ++ 123 +	7700	3	69	78	57	50	83	
Pina et al. [30]	812 197 ++	5000	4	77	75	72	40.5	91.9	Maximal copy
Heilmann et al. [31]	143 39 ++	7900			91	61			1 mm central for histology, rest for OSNA
Peg et al.* [32]	697 ++	15,000	1	70	76.7	55.2	41.1	85.5	TTL
Terrenato et al.	1140 172 ++ 146 +	2150	3	76	94.9	51.4	46.5	95.8	TTL
Nabais et al. [33]	598 58 ++	190,000	4	80	73.3	74.4		88.9	TTL
Espinosa-Bravo et al. * [34]	306 108 ++	120,000	1	71	47	85.3	56	80	TTL

List of the study analyzed (*: multicentric study; ++: macro-metastasis; +: micro-metastasis; AUC: area under receiver operating characteristics curve; Se: sensitivity; Sp: specificity; PPV: positive predictive value; NPV: negative predictive value; TTL: total tumor load).

## Data Availability

Not applicable.

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
