# Peer review of "A New Possible Cut-Off of Cytokeratin 19 mRNA Copy Number by OSNA in the Sentinel Node of Breast Cancer Patients to Avoid Unnecessary Axillary Dissection: A 10-Year Experience in a Tertiary Breast Unit"

_cancers, 2022, doi:10.3390/cancers14143384_

Round 1

Reviewer 1 Report

Comments:

1. The current manuscript is missing the details info on Table 1, such as patients' BMI, age, geographic info, etc.

2. On Table3, any statistical analysis on cut-off value? Without statistical analysis, it is difficult to know if it is meaningful for example 9.15 vs 7.7?

3. Any house-keeping mRNA compared to cytokeratin 19 mRNA in the current study?

4. On Table 1, estrogen receptor is alpha- or beta- or combined?

5. How many patients are triple negative status? how many are ER+ and PR+?

Author Response

We have read with particular attention the Reviewer's comments, which have prompted us to improve the quality of the manuscript and we sincerely thank the Reviewer for this opportunity.

Following, please, find the point-to-point responses which are enclosed into a .docx file attached.

1. The current manuscript is missing the details info on Table 1, such as patients' BMI, age, geographic info, etc.

We thank the reviewer for the suggestion. Actually, they are clinical details useful and consequently we have accordingly modified the Table 1. We added in the text that all the patients were European.

2. On Table3, any statistical analysis on cut-off value? Without statistical analysis, it is difficult to know if it is meaningful for example 9.15 vs 7.7?

We agree with the Reviewer, but Table 3 lists results of different ROC analyses from different studies. This means that different methodologies and different number of nodes were utilized to calculate the threshold and thus we have considered the data not comparable.

3. Any house-keeping mRNA compared to cytokeratin 19 mRNA in the current study?

The Reviewer’s question is very interesting and stimulating, but our answer, unfortunately, is negative. We have never validated an alternative house-keeping mRNA test to compare CK19 mRNA and that is why we didn't mention it in the current study.

4. On Table 1, estrogen receptor is alpha- or beta- or combined?

We didn’t specify that in the table and we thank the Reviewer for reminding us. We used alpha ER, Rabbit mAB, clone EP1, by Agilent. We have specified that and, accordingly, we have modified the table 1.

5. How many patients are triple negative status? how many are ER+ and PR+?

We have reported the data in the table 1, among the clinic-pathological characteristics. Anyway, following the Reviewer’s suggestion, we have added a sentence summarizing the immunohistochemical results of hormonal status and her2 status.

Hoping we have met the Reviewer's hints, we sincerely remain.

Reviewer 2 Report

In this manuscript, the authors have examined Sentinel lymph node (SLN) and Axillary lymph node status (ALN) in their breast cancer patients between January 2008 and December 2021. Their results indicated a new possible cut-off of CK19 mRNA copy number in SLN to predict cases at high risk of having a positive ALND. However, there are some remaining questions to be answered:

1, The authors identified a new cut-off (9150 copies) of all 328 SLN with macro- and micro- metastases (Line 165). However, The authors only examined the patients with macro-metastases (n = 202) for further ALND. Have the authors evaluated the patients with micro-metastases (n = 126) for ALND as well?

2, Could the authors include more details about their ROC analysis (Line 163-Line 166)?

3, What are the CK19 mRNA copies (Table 2)? Is the value 49.500 or 49,500?  

4, Some typo errors, Line 77, Line 160, SLN, not SNL.

Author Response

We thank the Reviewer for the inspiring comments useful to increase the quality of the manuscript and the interest of the readers.

Following, please, find the point-to-point responses which are enclosed in the attached .docx file.

1. The authors identified a new cut-off (9150 copies) of all 328 SLN with macro- and micro- metastases (Line 165). However, The authors only examined the patients with macro-metastases (n = 202) for further ALND. Have the authors evaluated the patients with micro-metastases (n = 126) for ALND as well?

We thank the Reviewer for this question. Actually, ALND was not perform in patients with micro-metastases. Only patients with macro-metastases were evaluated in this study, because in our Institution according to published data of IBCSG 23/01 trial (reff. 17 e 18) patients with micro-metastases were spared ALND.

2. Could the authors include more details about their ROC analysis (Line 163-Line 166)?

The Reviewer’s comment is meritorious and worth exploring. ROC analysis was further discussed in the test (2.3 Statistical analysis) adding a sentence with more details.

3. What are the CK19 mRNA copies (Table 2)? Is the value 49.500 or 49,500?

We are sorry for the typo and we have correctly reported the value in Table 2 and throughout the text.

4. Some typo errors, Line 77, Line 160, SLN, not SNL.

We thank once again the Reviewer for the attention of reading. Typos have been corrected.

Hoping we have met the Reviewer’s hints, we sincerely remain.

Round 2

Reviewer 1 Report

No more comments.